# Towards Best-Practice Healthcare for Transgender Patients: Quality Improvement in United Kingdom General Practice

**DOI:** 10.3390/healthcare13040353

**Published:** 2025-02-07

**Authors:** Carine Silver, Rebecca Calvey, Alexandra Martin, Joanne Butterworth

**Affiliations:** 1Rolle Medical Partnership, Exmouth Health Centre, Claremont Grove, Exmouth EX8 2JF, UK; 2St Thomas Medical Group, Cowick Street, Exeter EX4 1HJ, UK; 3Exeter Collaboration for Academic Primary Care, University of Exeter, Exeter EX4 4PY, UK

**Keywords:** transgender, gender dysphoria, general practice, primary care, quality improvement

## Abstract

**Introduction:** The ongoing care of transgender patients in United Kingdom (UK) general practice (GP) is hampered by a lack of UK primary care guidelines regarding the monitoring of treatments, despite the key role that general practice has in holistic lifelong care. This quality improvement project aimed to audit the monitoring of treatments and health screening in a GP practice population, across two large practices in southwest England, in order to drive local improvement and to identify gaps in wider healthcare support for this population. **Methods:** This project updated a previously published audit instrument, incorporating a novel, pragmatic standard, based on up-to-date UK gender clinic guidelines and the UK population screening programmes. National Health Service (NHS) Health Research Authority and Medical Research Council processes were used to confirm that this quality improvement project did not require formal ethics committee approval. An audit against this standard was performed for 176 transgender and gender-minority patients, to provide data on the consistency of the monitoring of gender hormonal treatments and reminders for appropriate population health screening programmes. **Results:** A total of 16% of those undergoing hormonal treatments had received optimal monitoring; 20% were missing the most basic hormone level monitoring. Reminders regarding appropriate health screening were rare in patients who had changed the gender markers on their electronic record. Long waiting lists, the use of private clinics, confusion around responsibilities shared between primary and secondary care and growing complex co-morbidity were demonstrated. **Conclusions:** This project supports previous calls for consistent evidence-based guidelines, improved data systems and adequately resourced primary and secondary care services to support the safe and effective lifelong care of transgender patients.

## 1. Introduction

The United Kingdom (UK) professional membership body, the Royal College of General Practitioners (RCGP) recognises that GPs face considerable challenges managing their transgender patient cohort, due to increasing demand, a changing medical and social landscape, over-stretched services and a lack of targeted training and education [1]. Survey data from the UK Government Equalities Office [2] suggested that a fifth of transgender patients had “specific needs that were ignored or not taken into account when they accessed, or tried to access, healthcare services” in the preceding 12 months. These issues are recognised internationally, with documented rises in numbers seeking gender interventions [3,4,5] and reported difficulties for transgender patients accessing care, whether for gender or general medical matters [6].

In the UK, patients usually first present to their GP in order to access referral to secondary specialist services at one of the six adult gender identity clinics (GICs) within the state-funded NHS (National Health Service), although some may use direct access private care, personally or insurance-funded. The situation for children is undergoing change: there are currently two children’s clinics, with six more envisaged. Long and growing waiting lists [7] for GICs put additional pressures on GPs, both to support the waiting patients and to commence hormonal treatments on ‘bridging prescriptions’ (also known as ‘harm-reduction’ prescribing). These pressures have been associated with a rise in referral and self-referral to private GICs. Patients move between private and NHS care in response to patient preference or service availability. If gender-affirming interventions are commenced in private care, a GP can agree to ‘shared care’, using a Shared Care Agreement (SCA), where roles and responsibilities for the prescribing, monitoring and review of patients are agreed between both primary and specialist sectors. Between NHS GICs and primary care, the situation is more complex; although the regulator, the General Medical Council (GMC), references the use of shared care agreements for transitioning between NHS secondary and primary care [8], the current NHS adult service specification [9] requires primary care to be involved in gender treatments before a patient is stably established (p. 17, [9]) and it does not mandate the use of SCAs. The latest RCGP transgender guidance [1] recognises ‘collaborative care’ between primary and secondary care, but the model it describes is inconsistent with the NHS England (NHSE) service specification. Unclear sharing of responsibility and re-referral mechanisms may also contribute to inconsistent care.

The first GP-level audit of transgender patient care in the UK was carried out across a sample of 20,136 patients in a single practice in 2021 [10]. It found no standardised UK guidelines for primary care and inconsistencies between clinic monitoring schedules, whether NHS or private (Table 1 and Table 2, ref. [10]). Since then, the UK gender medicine landscape has undergone changes: following the publication of the Cass review into children’s gender services [11], a recommendation to form a specific model of care for young adults (17–25 year olds) was accepted by the UK government [12]; a review of adult gender services was announced [13]; and there have been updates to international and local adult gender medicine guidance documents (described below). However, it remains the case that there is no UK guidance or quality standard available from the National Institute for Clinical Excellence (NICE) for England and Wales [14], from SIGN (covering Scotland) [15] or in Northern Ireland (which adopts NICE guidance [16] after endorsement).

Previous systematic and simple reviews have documented the degree to which gender-questioning adults [18,19] and children [20,21] exhibit high-levels of psycho-social and neurodevelopmental comorbidity. Whilst the reviews are largely international in scope, and focused on secondary care populations, a 2023 review of UK primary care survey data [22] showed similar trends of over-represented autistic spectrum conditions and mental health problems in trans patients, particularly younger age groups. Whilst GP training explicitly addresses caring for the whole patient (Domain 8 ‘medical complexity’ and Domain 12 ‘Holistic care’ [23]), Cass noted that ‘diagnostic overshadowing’ (becoming focused on a single aspect of presentation, at the expense of other underlying or contributory conditions) is an important risk when dealing with patients presenting with gender incongruence/dysphoria [11] (see below for details regarding language and terms). The new paediatric clinics are designed to incorporate a holistic multi-disciplinary team (MDT) approach to address wider needs [11]. Conversely, the referral pathway for adult GICs in the UK specifies the assessment and treatment of comorbidities prior to Gender Clinic referral (Appendix B, ref. [9]), potentially dividing the most complex patients’ needs between multiple services.

The RCGP advises that improvements in training are required for GPs in caring for their transgender patients [1]. The inconsistency and instability of a complex healthcare landscape may also increase the difficulty for GPs in staying up-to-date. Whilst uncertainty exists about the long-term effects of gender-affirming hormones (with or without prior gonadotrophin releasing hormone agonist (GnRH) use), harmful side-effects have been noted [24]. Therefore safety monitoring is a standard part of the lifelong care of transgender patients, paralleling the long-term monitoring of other medications, and patients with chronic health conditions. These rely on electronic systems of recall, although current IT systems do not work well for transgender patients [1].

Whilst the diagnoses of gender incongruence (ICD-11) and gender dysphoria (DSM-V) have different emphases see (p. 83, [11]), the current NHSE service specification [9] describes the process of gender dysphoria diagnosis at specialist ‘Gender dysphoria clinics’. For the purposes of this document, therefore, we have used the DSM-V term gender dysphoria, whilst accepting that not all patients with gender incongruence may suffer significant distress (dysphoria) related to their gender. Language in this area is acknowledged to be both sensitive and confusing; terms have been used in this document in line with the Cass Report [11].

The aims of this quality improvement project were to: determine an appropriate up-to-date audit standard for the safety monitoring of medical gender interventions; to use this standard to assess the quality of the primary care monitoring of existing gender-affirming endocrine treatments in a population-based sample of transgender patients and to better understand the psychosocial and neurodevelopmental complexity of this transgender population. The results of the project will be used to inform a robust plan of local improvements to transgender patient primary care and to highlight current gaps in wider healthcare support needed for this population.

## 2. Materials and Methods

### 2.1. Setting

This multi-centre quality improvement project was carried out across two general practices in Southwest England. The audit project consisted of a general practice population census, review of electronic health records (EHRs) and audit against a standard developed for this project, against a background of patient demographics and transgender care pathways.

The two practices were chosen for their joint interest in improving care for their transgender patients and large combined population of over 73,000 patients (see Table 1). It was anticipated that, due to different demographics and approach to the care of transgender patients, a comparison of audit results between the two practices may provide tentative suggestions on how pathways and local services could be improved. Practice 1 has a large population of adolescents and young adults, hosts a university student health service, has good informal links with the local adult GIC, and holds regular transgender patient clinics, run by GPs with a special interest in gender medicine. Practice 2 serves an older population and has no clinicians with a special interest. Both practices collaborate with NHS GICs and, on a case-by-case basis (after quality assurance), with private GICs. Both practices were rated ‘Outstanding’ by the Care Quality Commission in their most recent inspection of services.

### 2.2. Audit Standards

Difficulties in finding an appropriate audit standard have been previously described by Boyd et al. [10] who provided a detailed comparison of available guidance documents from national, regional and UK sources. For their audit work, they used the secondary care Endocrine Society Guidelines [25]. However, as updates to previous guidance documents have become available, this has become out of date. In order to identify an appropriate, evidence-based audit standard, searches were made using PubMED and Google for up-to-date national and international medical guidelines pertaining to gender interventions. Specific searches were made for updated versions of the guidelines referenced by Boyd et al. [10], and references therein were further interrogated.

Population health screening programmes, where available, are decided by national healthcare bodies, based on population characteristics and health needs. The UK has national programmes which differ slightly between the devolved nations, overseen by the UK National Screening Committee (UKNSC) [26], and these recommendations were used as the basis for audit in this project.

### 2.3. Instrument Update and Use

The previous data collection spreadsheet instrument [10] was obtained and the tool updated with our chosen audit standards for hormone monitoring and health screening (detailed further below). Continuous quantitative data were recorded for age, the date of presentation to gender services, referral date and the length of time on treatment; discrete quantitative data were collected regarding the number of appointments with clinics; and nominal data were collected in all other categories, using drop-down options for ease of analysis. Some free-text boxes were used to capture themes (e.g., clinic name, complications from surgeries; reasons for ceasing hormone treatment and adverse childhood experiences). The new tool was piloted by three clinicians (ten patients from Practice 1 (RC) and twenty patients from Practice 2 (CS and AM)) to determine ease of use, the organization of data and terminology; nominal options for chosen gender identity were updated in line with patient preference as noted in the records. After team agreement, the final version was used to record data across both practices: a blank version of the final instrument is available from the corresponding author.

### 2.4. Patients/Sampling Frame

There was no pre-determined sample size, as this quality improvement project was not designed for statistical power or significance. Gender questioning and transgender patients were identified by consistent search terms across both practices’ electronic health record (EHR, SystmOne software). All patients with current or historic gender-related diagnostic codes (for the full list, see Appendix A) prior to or on the study date were included in the census. There were no exclusions on the basis of age; the status of referral to secondary gender services; the status of existing or planned affirmation (medical, surgical, both or neither); whether or not the patient had applied for a Gender Recognition Certificate or whether the patient had changed their EHR ‘gender marker’. No comorbidities were used as a basis for exclusion, with the exception of one patient who had a comorbid Variation of Sexual Development.

Due to the process of NHS ‘gender marker’ change, which may also involve the redaction of notes to delete references to birth-registered sex, gender-associated coding may be lost and patients missed when using only coding as a basis for search. Therefore, an additional search was made for patients currently prescribed endocrine treatments for gender-affirmation (see Appendix A).

The population census was carried out across both practices on 9 January 2024; only medical history prior to this date were used to populate the audit instrument. All records were manually checked for appropriateness.

### 2.5. Data Extraction

Three clinicians extracted data, two from each practice (CS/RC and CS/AM). Double data collection was performed for eighteen patients to check consistency in data collection between clinicians. Only events or data from before the census date were recorded. Data were stored in protected folders locally within the practices and anonymised after collection. Data items included demographics (age, race, birth-registered sex, gender identity and psychiatric and neurodevelopmental co-morbidities); age at (and the date of) medical presentation with gender dysphoria; pathway and timeline to gender treatment; the type of interventions; complications and medication monitoring and population health screening against our audit standard. Clinic letters and shared care agreements were documented: a letter was considered adequate if detailed diagnosis, assessment, prescription required and monitoring recommendations were all present.

### 2.6. Strategy for Data and Statistical Analysis

Simple frequencies were calculated, and tentative trends are presented descriptively. As this study is quality improvement work, no tests for statistical significance were used. SAGER guidelines [27] have been followed to avoid the conflation of sex and gender identity and to support adequate the disaggregation of data. Gender identity is of primary importance to the patient, but as medical treatment (masculinising or feminising gender-affirming interventions), and its associated monitoring and potential side-effects, is driven by birth-registered sex, data have been disaggregated by treatment given or by sex.

Missing data were common in all categories other than basic demographic data, and are detailed in the narrative and tables. For referral to GIC, if only year was known, date was defaulted to 01/01 of that year.

### 2.7. Ethical Considerations

The joint NHS Health Research Authority and Medical Research Council process was used to determine and confirm that the project was quality improvement, thus not requiring formal research ethics committee approval [28]. Due to the relatively small number of patients involved, data are aggregated across broad categories and both practices for presentation.

## 3. Results

### 3.1. Choice of Audit Standard

Prescribing, monitoring and health screening guidance from the identified updated guidance documents was collated and is tabulated in Appendix B (Table A1 and Table A2). This study found that monitoring and screening advice varied substantially between clinics, even within the UK. As a result, the team agreed to develop its own audit standard, based on a consensus approach.

Whilst there is uniformity between the formulations of hormones used for gender interventions between UK clinics, alternative formulations and laboratory reference ranges at clinics overseas limit the relevance of their guidance to the UK setting. Therefore, specific blood monitoring recommended by at least three UK clinics was included in the audit standard, and annual testing was preferred over 6 monthly or 2 yearly for practical recall purposes. Details of national and international guidance, and the variability of recommendations, are further discussed in Section 4.1.

Our audit standard is presented in Table 2. A panel of yearly hormone monitoring is widely recommended, in addition to blood pressure and weight (or body mass index, BMI) which are also commonly recommended due to the potential metabolic effects of gender interventions. The need for pelvic ultrasound, to identify endometrial thickening as a result of testosterone supplementation, whilst variably advised by clinics, is included in our audit standard.

Whilst screening advice for transgender people in the UK is available [29], automatic screening invitations are based on electronic health record (EHR) gender markers and so the responsibility for anatomy and sex-appropriate screening lies with the patient and with the GP surgery. Transgender patients who change their gender markers to better align with their gender identity are therefore are at risk of being lost to anatomy-appropriate screening programmes. There are also uncertainties related to specific screening needs in patients undergoing gender hormonal interventions (discussed further in Section 4.1). As some population screening programmes are applicable only many years after transition, a reliance on patients or clinicians being aware of the continued relevance of programmes—which may additionally have changed in the intervening years—is inadequate.

Our audit standard proposes that screening reminders should be available within the notes of patients who had changed their gender markers, as per birth-registered sex and if relevant to their anatomy: for example, a cervical screening reminder for a birth-registered female (BRF) patient with an ‘M’ gender marker, unless having undergone hysterectomy. In addition, patients who were taking hormones but who had not changed their gender markers should have screening reminders relevant to their risk of developing disease: for instance, a birth-registered male (BRM) patient on feminizing hormones ought to be offered abdominal aortic aneurysm (AAA) screening, but would not be automatically included in the NHS programme due to an ‘M’ gender marker. Finally, a reminder of the ongoing presence of a prostate in BRM patients with an F gender marker was included in our standard, due to the risk of overlooking the ongoing presence of a prostate in future differential diagnoses. Although the need for a dual energy X-ray absorptiometry scan (DEXA, bone density scan) is dependent on patient history, it is important that the need is considered in each patient. Our audit standard considered whether a DEXA had been explicitly considered for each patient.

**Table 2 healthcare-13-00353-t002:** Audit standard developed for this project.

	Feminisation Therapy for Birth- Registered Males (BRM)	Masculinisation Therapy for Birth- Registered Females (BRF)
Blood test monitoring (annual)	Oestrogen, testosterone, lipids, LFTs, prolactin	Testosterone, FBC/HCT, LFTs, lipids and HbA1C/fasting glucose
Other monitoring	BMI and BP (annual)	BMI and BP (annual) Pelvic USS unless hysterectomy (every 2 years)
Population health screening	As per NHS programme for BRM plus breast screening as per NHS programme for BRFs. Explicit consideration of DEXA scanning recorded	As per NHS programme for BRF unless post-mastectomy/hysterectomy Explicit consideration of DEXA scanning recorded

FBC: full blood count; HCT: haematocrit; LFTs: liver function tests; BMI: body-mass index; BP: blood pressure; USS: ultrasound scan; DEXA: dual energy X-ray absorptiometry scan (bone density scan).

### 3.2. Population Census and Pathways

#### 3.2.1. Demographics

Basic demographic data of patients in the patient population census are shown in Table 3.

The mean and median age of all patients at first recorded presentation to medical services with gender issues was 21 and 19 years. Birth-registered females presented at a mean age four years younger than birth-registered males, with 56% before the age of 18 and 28% between 18 and 25 versus 27% and 48%, respectively.

The majority of patients identified as trans (72%, of which 35% were trans man and 35% were trans woman) or non-binary (18%, of which 9% were non-binary Transmasc and 7% were non-binary Transfemme) at presentation. The remainder were either unsure (4%) or other (non-binary 2% and agender/genderfluid 1%). Data were missing for 3% of patients.

Ethnicity data were missing in 21% of cases. Recorded ethnicity was majority White British (40%) or British or Mixed British (31%). Other White (4%) and other Mixed or Non-White ethnicities (4%) were recorded. This compares with a (non-age-matched) total practice population average which is 93% White British (see Table 1).

First presentations to medical services of patients with a gender-related issue or explicitly documented transgender or gender minority identity are shown by year in Figure 1, with the vast majority presenting in the decade 2014–2023. The mean number of years since the first recording of gender concerns, dysphoria or a changed identity was 5 years, with a median of 3 years (for BRF, 4.7 years and 4 years and for BRM, 5.4 years and 3 years). Overall, there has been a 5-fold increase in presentation with transgender or gender minority identity in the past decade.

Mental health and neurodevelopmental diagnoses were common (see Table 4). In addition to those tabulated, other mental health diagnoses, or presentations not meeting the threshold for diagnosis, included post-traumatic stress disorder (PTSD) or complex PTSD (n = 10), hallucinations not meeting threshold for psychosis (n = 8), verbal and/or motor tics (n = 5), emotional dysregulation (n = 6, of which two had documented violence and aggression), trichotillomania (n = 3) and paranoid schizophrenia (n = 2). Selective mutism was noted in two patients. Adverse childhood experiences were recorded in 75/176 (43%) patients, including 37 of the 75 (49%) reporting being bullied, 10 (13%) reporting domestic violence and 16 (21%) reporting sexual abuse or assault. BRFs were over-represented in recorded mood disorders, self-harm, eating disorders, adverse childhood experiences and previous care from secondary mental health services.

Amongst those using substances, smoking was the most common (22%, 35/159), 14% of patients (21/151) engaged in ‘other’ drug use and 12% (18/153) had current or past hazardous alcohol use (denominator representing patients for whom data were available in each category).

#### 3.2.2. Clinic Referrals and Pathways

Referral pathways and current clinic status, including uncertainty in current clinic status due to missing letters, are shown in Figure 2. The total number of patients that were seen by either a private or NHS clinic was 96 (63% of all patients referred to a GIC, or 54% of all patients recorded as transgender patients in our census).

For those patients currently on the NHS GIC clinic waiting list (see top right of Figure 2), the mean waiting time is 2.2 years (range 0 to 8 years, see Figure 3), with 87 patients (49% of the transgender patient population) at the time of the study currently on a waiting list for NHS gender care. For 6/87 patients, referral date was entirely missing. Waiting time for private clinics was not recorded.

A total of 18 clinics (six NHS and twelve private) were recorded as having seen or treated patients. Two patients had attended an overseas clinic and nine had no recorded clinic name.

Self-referral had resulted in NHS clinic treatment in one case, and private treatment in 62. In two of these, a GIC requested additional GP information before recommending treatment. It was unclear how the referral had occurred due to lack of letters in 15 cases.

The total number of appointments offered by clinics before recommending medication, and who they were with, was often missing from communications. For the purposes of this analysis, if a clinic letter mentioned at least two appointments, it was assumed that there were only two. Thus, the mean number of appointments, for 40 patients where data were available, was 4.3 (range 1 to >20). The average number of appointments offered to adults prior to treatment is 2.6 NHS (4.6 if the more extended Gender Identity Development Service—GIDS, the former UK paediatric gender clinic—assessments are included) and 1.8 private.

Figure 3 summarises the key parts of the medical and surgical pathways of the patients in this census; whilst 96 patients are recorded as having been seen in GIC, and 96 patients are recorded as having ever undergone endocrine treatment, these are not all the same patients, due to patients having started hormones via other routes (such as unregulated medications) and patients having ceased taking hormones. Only two of the 96 patients seen in a GIC clinic (private or NHS) had been discharged without being recommended or prescribed hormonal medication: one was seeking surgical treatment only, and the other was discharged from GIDS prior to medication being commenced as an adult.

Adequate letters were found in the notes for 76% (25/33) of patients last seen in an NHS clinic and 44% (23/52) seen in a private clinic (see Table 5). In 16% (14/85) of patients, the GP prescribed cross-sex hormones despite a lack of adequate letters from GICs in the notes.

Six patients had undergone gamete preservation (five currently on hormonal medication; five BRM and one BRF), whilst 44/96 (46%) patients who were currently or previously taking medication had not elected to preserve gametes. Data were missing for the remaining 47 (49%).

The initiation of medication is shown in Figure 3. In 32% (31/96 cases), the GP had taken over the prescribing after GIC assessment, and prescribing was initiated by a GP at the request of the GIC in a further 38 (40%). In nine (9%), patients had started medication themselves through unofficial channels; in seven of these nine a GP took over the prescribing. In a further seven (7%) cases, a GP started the medication under ‘harm-reduction’ prescribing, although in one case, prescribing by the GP was later stopped after private clinic assessment. Data on initial prescription were missing for 12 (13%) patients, but a GP had taken over the prescribing for 10 of them.

#### 3.2.3. Medical and Surgical Management

Ninety six of the 174 total patients (55%) had ever received hormonal or blocker medication (8 missing data). Of the 96 who were documented as having started medication, 87 (91%) continued taking concurrent medication, 6 stopped and 3 were unknown). Details are shown in Table 6 along with types of medication prescribed. For the eight patients who had received puberty blockers prior to receiving cross-sex hormones, the average time before introduction of cross-sex hormones was 20.7 months (n = 6; missing data 2).

In a further 5 patients, data were missing such that it was unclear whether medication had ever been taken. Thus, at the time of the audit 87 patients were currently recorded as being prescribed cross-sex hormones (38 BRF, 49 BRM).

The census found 40 patients had undergone some form of gender-affirming surgery (42% of the 96 who had been seen by a GIC, or 23% of the total 176). The types and number of surgical procedures recorded are shown in Figure 3. One patient who had a double mastectomy for a BRCA mutation was not included as this was not for a gender-affirming indication. Surgery was a mixture of NHS and privately-funded surgery.

#### 3.2.4. Undesired Outcomes

Problems had been recorded in seven of the 96 patients (7%) who had taken hormones/puberty blockers (five BRF and two BRM), diagnosed as having been caused by the hormonal treatments (uro-gynaecological problems including vaginal atrophy requiring topical oestrogen, abdominal cramps, dyspareunia, post-coital bleeding, and vaginal infections including abscess; hot flushes; unwanted male pattern baldness and osteopenia/osteoporosis). A further 24 (25%, 19 BRF and 5 BRM) had documented issues which were considered likely (but not specialist-confirmed) to be medication-related including worsened mental health; elevated blood parameters of testosterone, oestrogen, prolactin, HCT, Hb or cholesterol; uro-gynaecological problems and acne.

Surgical complications requiring further early surgical or GP care (such as infection or poor wound healing) were described in 6/37 (16%) patients who had had surgical procedures, whilst a further three (8%) patients had considerable longer-term surgical complications (requirement for revision surgery). There were 15 (41%) patients for whom data on possible surgical complications was missing.

Of the six patients (16%) who had stopped taking cross-sex hormone medication, the GP informed the GIC in four cases. In one, the GIC contacted the GP with a plan regarding stopping medication. Reasons recorded for stopping medication were a change in self-identification (two), side-effects from medications (one) and medication supply issues (one). For two patients, no reasons were recorded in the notes. The average time of taking medications before stopping, where data were available, was 18.5 months (n = 4, two were missing data).

### 3.3. Audit of Monitoring Against Standards

#### 3.3.1. Monitoring of Hormonal Interventions

The monitoring of gender treatments in patients known to be taking hormone therapy (even where the formulation of medication was not recorded) against individual parameters is shown in Table 7. In addition to the data tabulated, of those pre-surgery trans men (BRF) who had been taking testosterone for >2 years (n = 21), five had had an endometrial scan and 16 were either overdue for endometrial scanning, or information was missing (due to missing letters or dates).

#### 3.3.2. Communications

There was a large proportion of patients on an NHS GIC waiting list (64% of all of those referred to NHS clinics) and a high degree of private care accessed by patients (68% of all patients seen in a clinic). Communications with GPs were often missing or of poor quality. For patients who had commenced treatment with a GIC, 20% (NHS) and 60% (private) had neither a formal discharge letter nor a letter dated from within the preceding 12 months. Adequate letters were missing in 14% of NHS-treated patients and 48% of those under private care. Despite this, prescribing was taking place in primary care.

#### 3.3.3. Population Health Screening

Less than a fifth of patients who had changed their gender markers had appropriate EHR reminders for recommended population health screening noted over that to which a patient would be automatically invited (Table 8). No patients had records showing an explicit consideration of the need for DEXA scanning although 3/8 (one BRM and two BRF) of those considered at risk of osteoporosis had undergone DEXA scanning. Risk factors included long puberty blockade, long periods of suboptimal hormone treatment or DEXA advised by GIC but not carried out due to the rejection of referral by the receiving radiology service. Whilst not part of our audit standard due to unclear evidential support (Section 4.1), 0/19 (0%) BRM patients taking feminising hormones but with a male gender marker had had breast screening explicitly considered in their notes.

#### 3.3.4. Differences Between Practices

Demographic data shown in Table 3 show a higher prevalence of transgender patients at Practice 1 than Practice 2. The proportion of birth-registered sex and age profile of the transgender population is similar between both practices, particularly age at presentation. The number of patients who have changed their gender marker is lower at Practice 1, consistently across both sexes.

The proportion of patients with adequate letters from GICs was lower at Practice 1 (54% compared to 63% at Practice 2) with a greater proportion of adequate letters from NHS clinics at both practices. For Practice 1, 73% of patients where prescribing had been taken over by GP had an adequate letter, compared with 85% at Practice 2.

For monitoring, though numbers were poor for both, there was a trend towards more consistent monitoring at Practice 1: 20% of the masculinising and 16% feminising patients were fully monitored, compared to 13% and 8% at Practice 2. The difference was more marked for blood-specific monitoring: 50% of the masculinising and 76% of feminising patients had full blood monitoring at Practice 1, compared to 38% and 25%, respectively, at Practice 2.

The proportion of patients with EHR screening reminders was higher at Practice 1 for breast and AAA screening (11% and 4%, compared with 0% for both at Practice 2), but higher at Practice 2 for cervical and prostate reminders (38% and 22%, respectively, versus 15% and 9%, respectively, at Practice 1).

## 4. Discussion

### 4.1. Guidance Documents and Audit Standard

NICE and SIGN are the two UK bodies who publish trusted, evidence-based guidelines for UK clinicians across most areas of medicine but as earlier stated there are no NICE or SIGN guidelines for gender interventions, aimed at either primary or secondary care, and to the best of our knowledge, no such guidance is in preparation.

As shared care in the UK applies almost exclusively to adult patients, we limited our search to adult gender medicine guidance (although some documents address the care of both children and adults). We included all of the UK gender clinic guidelines, since primary care monitoring in UK general practice is generally based on advice from one or more of these clinics. Other international settings vary in their use of national healthcare guidelines and locally adapted versions can be found. For example, a recent study [31] auditing gender hormone initiation amongst US transgender veterans references its own Veteran’s Health Administration (VHA) guidance, which is acknowledged to vary from other widely-cited US guidance [25,32]. Due to the unclear developmental basis of such locally adapted guidelines, we restricted our comparison work to the most widely cited international English-language guidelines.

In particular, we identified updated guidance from the World Professional Association for Transgender Health (WPATH) [32], AusPATH [33], NHS Scotland [34], NHS Wales [35] and the NHS Gender Identity Clinics (GICs) at Sheffield [36], Nottingham [37] and Charing Cross/Tavistock [38]. The two remaining NHS GICs (Exeter/The Laurels [39] and Leeds [40]) had not published updates but their existing guidance was included in the comparison. Three private clinics operating in the UK and frequently accessed by our patient population (GenderCare [41], Gender Hormone Clinic [42] and GenderGP [43]) responded to email requests for monitoring guidance.

We also included the University of California guideline [44], as the only international guideline aimed at primary care. The Endocrine Society guidelines [25], whilst now one of the older documents, have been heavily influential in the development of later guidance (p. 130, [11]), and were also used as the audit standard by Boyd et al. [10] and therefore have been retained for comparison. The European Society of Sexual Medicine Position Statement [45] was previously excluded [10] due to a lack of detailed monitoring protocols, but is included here as the only available pan-European guidance. The guideline appraisal carried out for Cass concluded that only two paediatric guidelines (from Sweden and Finland) were of sufficient quality to be recommended for use in practice (p. 130, [11]); the English-language summary of the Finnish COHERE guidance for the treatment of transgender adults [46] lacks practical details on safety monitoring and has therefore not been included in our comparison, and Swedish adult guidance was found. A link to a guideline by the Irish College of General Practitioners regarding GP care for transgender health [47], available online but through their library, was not included due to a lack of detailed monitoring guidance. Similarly, the guidance provided by GenderGP was excluded as it did not provide enough detail to contribute to the development of our audit standard. Finally, Red Whale [48] guidance (a widely used primary care CPD tool in the UK) no longer appears available in their library.

It was outside of the scope of this project to assess the quality and methodological rigour of the guidelines available. However, it is noted that the quality of development and underpinning evidence is criticised in the literature: Dahlen et al. [49] analysed international transgender clinical practice guidelines (CPGs) in a systematic review, finding them low quality. Ziegler et al., in their systematic appraisal of international CPGs relevant to the primary care of paediatric [50] and adult [51] transgender patients scored all CPGs poorly for developmental rigour. In addition, the Cass report ([11] chapter 9) criticised international guideline creation: a circularity of transgender medicine guideline development was mapped, with major guidelines referencing one another, thereby creating the illusion of consensus despite the poor quality of the underpinning evidence. Dissimilarities between other healthcare and regulatory settings and that of the UK may limit the applicability of guidelines; for instance, the AusPATH guidance [33] is based on the model of ‘informed consent’, rather than a diagnostic model defined in the UK gender service specification [9]. Thus, our audit standard, derived for a UK model of healthcare, patient population and intervention regimes may not be applicable to wider settings, and by limiting our search to English-language documents we may exclude valuable guidance developed elsewhere.

However, our detailed comparison between guidelines highlights some of the areas of inconsistency, driven by the paucity of the evidence base (p. 6, [38]), that could be usefully addressed by wider (including international) research of the outcomes of gender interventions. For masculinisation therapy, there is wide agreement that testosterone levels and full blood count (FBC) require regular monitoring, in order to ensure haematocrit (HCT) remains within safe levels. The frequency of recommended monitoring varies from 4–6 monthly [37] to 6–12 monthly [25,32,45]. There is no consensus on a safe upper limit of HCT (from 0.48 L/L [38] to 0.54 L/L [40]) nor action to be taken by the GP if out of range, with guidance varying from specialist advice [35,40] or urgent referral to a haematology service [34,39] triggered by various upper maximum values, to no advice at all [25,32,36]. Other commonly, but not uniformly, advised monitoring includes liver function [33,36,37,38,39] and lipid monitoring [33,35,36,37,39,42]. Blood pressure and weight (or body-mass index, BMI) monitoring are widely recommended, from regular intervals [32] or annually [36,38] to every 2 years. Measuring fasting glucose or glycated haemoglobin (HbA1C) is recommended by a minority of clinics [33,36,39]. Other irregularly mentioned tests are oestradiol [33,36,39], luteinising hormone [33], renal function [33,36], prolactin [33,36], thyroid function [36] and calcium [39]. Finally, pelvic US screening every two years is recommended (for patients who have not had a hysterectomy) by three UK clinics [35,36,38].

For feminising therapy, measuring oestradiol is widely recommended, though the timing and frequency of monitoring is not always explicit [32,44], or consistent: every 2 years [34], annually [37,38,40] or every 6 months [33,34,35,39]. The target range for oestradiol is variable (e.g., 250–1000 pmol/L [33], 200–600 pmol/L [34,39] and 400–600 pmol/L [37,38]). In addition, monitoring testosterone levels is widely recommended (with frequencies varying from every 3 months [25] to every 2 years [39], and target ranges from <1.7 nmol/L [25] to <3 [38,39]). Target range was missing from two UK guidance documents [35,37]. Other blood test monitoring recommended (at intervals up to every two years) are liver function [33,34,35,36,37,38,39,40], lipids [33,34,35,36,37,39,40,44,45], prolactin [34,35,36,37,38,40] and glucose/HbA1C [33,36,39]. Weight/BMI [33,34,35,36,37,38,39] and blood pressure [33,34,36,37,38,39,40] monitoring are commonly recommended whilst renal function [33,36], thyroid function [36] and follicle-stimulating hormone/luteinising hormone [33,37] were recommended by a minority of clinics. There were some changes noted between current and previous versions of guidance for feminising therapy, such as no longer recommending blood pressure, BMI, lipids or HbA1C [35], or newly recommending lipids, BMI, BP and testosterone levels, but not prolactin [34].

Several uncertainties in the need for health screening were noted. Firstly, many clinics (and population health screening guidance aimed specifically at transgender populations [29]) recommend breast screening for patients undergoing feminisation therapy, as the risk of breast cancer is thought to be increased, although not as much as in birth-registered females [52,53]. Patients undergoing masculinisation therapies are known to be at a higher risk of breast cancer than birth-registered males [26,29] yet no guidance about screening by mammography is given, although one document recommended regular peri-areolar breast examinations for post-mastectomy patients [25]. Secondly, there may be an increased risk of osteopenia in patients who have had breaks in exogenous/endogenous sex steroids (e.g., long puberty blockade, or absent/insufficient hormonal replacement post-gonadectomy). Thus, in some cases, bone density scans may be of benefit. However, where explicitly addressed in UK clinic guidance, the length of break triggering the need for a scan varies from every 6 months [34] to every 2 years [36]. Thirdly, two clinics [35,36] recommend that BRM patients undergoing feminisation therapy should be included in the abdominal aortic aneurysm (AAA) screening programme. The risk for BRF patients undergoing long-term masculinisation is described as ‘lower’; these patients are invited to choose [34,36] whether or not to take part in the screening programme, in line with UK government transgender guidance [29] although not covered by National Screening Committee recommendations [54]. The need for screening in patients who underwent puberty blockade followed by cross-sex hormones is not described. Finally, whilst there is no prostate cancer screening programme in the UK, the ongoing presence of a prostate may be relevant to future health needs and was included in our audit standard.

Whilst there are clear limitations in our limited, consensus-driven approach (such as the amplification of the illusory concordance as noted by Cass above), our audit standard forms a pragmatic tool against which to judge UK primary care monitoring, until such time that national peer-reviewed, standardized, evidence-based guidelines are published.

### 4.2. Demographic Findings

Whilst formal comparisons were not the aim of the project, our findings are consistent with other UK studies. A large 2023 UK primary care population survey [22] described a transgender population that was on average young, more likely to be from an ethnic minority or a deprived area of the UK, with higher levels of mental health and neurodevelopmental concerns at a younger age. A recently published review of primary care records of children [55] shows a substantial recent increase in the prevalence of children and adolescents with gender dysphoria, although no link to social deprivation was found. Our audit also found a young, complex transgender population. The age profile and prevalence, which was markedly different between the two practices (Table 1), might relate to Practice 1 having a higher proportion of adolescents and young adults registered. The noted increase over the past decade in the presentation of transgender or gender minority patients seen in adults [3], young people [21] and primary care populations [55,56] is supported by our data.

Our study found evenly split groups (see Table 1) where other reviews have found trends towards more birth-registered female presentation [5,10,21,56], male predominance [3], or increasingly even split [4]. Causes of variation are not clear and may rely on the specific context of the studied population (e.g., primary versus secondary care, timing and region). Our data support a trend towards a younger average age of presentation of BRF patients than BRM, echoing the female predominance seen in paediatric populations [11]. Data collection for transgender patients has been shown to be frequently suboptimal; an international systematic review [57] found that 29% of studies did not specify how gender identity was ascertained, and data endpoints were not disaggregated by gender identity or birth-registered sex in 35% (as high as 65% for non-binary patients). In this census, a fifth of patients identified outside the gender binary of trans man/trans woman. This is in line with UK census data from 2021 (since designated as ‘in development’) suggesting 20% of those stating a gender identity not in line with birth-registered sex identified as non-binary [58]. The numbers should be treated with caution as they may be incorrect due to difficulties with participants understanding the survey questions [59]. The survey data of LGBTQ+ youth from the Trevor Project in 2021 [60] reported 26% of its sample identified as non-binary (of whom 50% thought of themselves as transgender) with a further 20% ‘possibly’ non-binary; it does not cite how many were seeking medical treatment. By contrast, Boyd et al. [10] found only 4% of their transgender and gender minority cohort were non-binary. Reasons may be related to methodological differences and surveying different parts of the pathway, or possibly due to changes in trends of self-identification over time. However, the number of non-binary patients in our census is important, as it is unclear how this population is best served within the current paradigm of gender interventions.

Our data show a high proportion of patients with comorbid anxiety/depression (74%), along with high rates of other mental health diagnoses (including 27% with recorded substance misuse, 16% with ASD and 13% with ADHD, a further 14% awaiting ASD/ADHD assessment and 6% with PTSD/cPTSD). There was a high degree of secondary care psychiatric support, and 43% with documented adverse childhood experience. Whilst this project does not compare with background rates of psychosocial and neurodevelopmental conditions, our data are consistent with the high rates of psychological and neurodevelopmental comorbidities seen in other studied populations [18,19,20,21,22]. A large recent systematic review of international data [57] supports these trends, though it cautions that heterogeneity in populations and ascertainment methods make comparisons of rates difficult. In addition, high and middle income countries are over-represented in existing data, limiting global transferability.

Our ethnicity findings do not rely on patient self-report, but there is a high proportion of missing data (see Section 3.2.1), and our transgender cohort may not represent the ethnic mix of the whole population due to the influence of a (diverse) university student community. Comparing ethnicity data is particularly difficult as the findings of the over-representation of ethnic minorities within the transgender population [22], and the 2021 census [58] have been challenged [59,61]: data regarding ethnicity may be compromised by questions regarding sex and gender being confusing to non-native speakers of English.

Detransition is differently defined, contentious and difficult to estimate. Based on stopping hormones, this audit found a rate of between 6% and 9%. This is in line with a published GIC case note review [62] but given that detransition may take years [63,64] is likely to be an underestimate of ultimate prevalence given a young, mostly recently transitioned or currently waiting cohort. In response to the Cass report, NHSE has announced [12] first steps towards a detransition pathway which may lend support to GPs caring for this poorly understood and underserved population.

### 4.3. Main Audit Findings

In line with Boyd et al. [10], we found conspicuous deficiencies in the routine monitoring of gender interventions. We found that a substantial proportion of relevant patients had not had the most basic monitoring of oestrogen (27%) or testosterone levels (11%) and less than one in five transgender patients had had optimal annual blood monitoring. In general, masculinising therapy was better monitored, but annual BP and BMI, which were recommended by most UK clinics, were absent in about half of both populations. Whilst it is likely that the lack of optimal monitoring (such as low numbers of patients having blood monitoring of prolactin or HbA1C, or pelvic ultrasound every 2 years) may stem from inconsistent clinic monitoring guidance and a lack of national GP-focused guidelines, the large number of patients lacking basic hormone or BP/BMI suggests improvements are needed in local practice recall systems.

Practice 1 demonstrated better monitoring than Practice 2, particularly for blood monitoring in feminising patients. This would suggest that having a clinician with some focus on gender medicine was somewhat helpful in ensuring regular blood monitoring.

Population health monitoring needs were also poorly recorded, with cervical cancer screening better documented, particularly at Practice 2. It may be that cervical cancer screening is of more imminent concern due to the younger entry age onto the screening programme, unlike the breast or AAA screening programmes and prostate problems which do not affect patients until much older.

The examination of patient pathways demonstrates the complexity of routes taken with gender-affirming interventions sought from a large number of different NHS, private and unregulated sources. The average number of appointments at private clinics (1.8) is below the NHS specification minimum of two [9], whilst almost no patients were discharged without gender interventions being recommended. A substantial proportion of patients were shown to be receiving gender intervention from their GP without adequate clinic letters, and a number of patients had not been assessed at all by a GIC, with the GP initiating prescriptions. There is inconsistency within, as well as between, practices as to whether the safety monitoring of unregulated hormones is taken on, or bridging prescriptions started elsewhere are continued. The confusion and complexity of this landscape is likely contributory to the poor safety monitoring seen in our data.

### 4.4. Strengths and Limitations

Strengths include a natural population census and large sample size, with the manual retrieval of data ensuring comprehensive, detailed and accurate descriptions with cross-checking between data collectors to ensure consistency. Whilst only EHR data were accessed, full notes retrieval allowed a detailed understanding of population characteristics and pathways, increasing accuracy. The capture of both gender identity and birth-registered sex (the ‘two-step’ method [57]), along with data disaggregation, allows for full transparency of results. Despite this, it is recognised that coding inconsistencies or omissions, missing letters due to patient moves between practices or redacted data resulted in difficulties in identifying transgender and gender minority patients, which may have resulted in the omission of some patients from the census and the overlooking of important history (e.g., psychiatric or social history). No early records were available for patients who moved into the practices from overseas or from devolved nations that operate a different EHR, although occasionally information existed in more recent letters. Patients with transgender identity who had not presented to their GP due to not wishing to medically transition, who were accessing unregulated hormones or who were purely under the care of private clinics would also be absent from the census. Such missing data may also cause some results to be underestimated.

A further limitation is the low prevalence of transgender patients, resulting in a relatively small sample size, and a population sample from a rural area in the UK with low deprivation and little ethnic diversity. Whilst generalising to other settings was never an aim of this project, the transparency of results may allow some transferability to similar populations, both nationally and internationally, in light of the common trends and themes described [10]. The limitations of our developed audit standard are detailed in Section 4.1; whilst relevant to the UK healthcare environment, it may be of limited international relevance. However, the evidential gaps in requirements for safety monitoring are of international importance.

Reasons for not seeking GIC referral, for not attending initial appointments and whether or not patients on waiting lists are still seeking transition could not be established within this study. As GIC referral dates were rounded to 01/01 if only the year of referral was available, waiting list times may be slightly overestimated.

The risk of subjectivity in judging clinic letters for adequacy and the relevance of patient history was addressed through inter-clinician QC and practicing reflexivity [65] at the outset of the project. Despite this, and the range in author experience and outlook, it is acknowledged that a certain degree of bias will be irreducible.

### 4.5. Implications for Patients, Clinicians, and Policy Makers

The growing number of patients has important implications in terms of primary care resources. This extends beyond the need for gender services into general mental health services and neurodevelopmental assessment and support. The 14% of our total sample who were waiting for ASD and/or ADHD assessments (Table 4), suggests a logjam in wider mental health services, not merely GICs. Unmet mental health needs make diagnostic overshadowing of greater concern, and require GPs to heed the RCGP’s [1] advice regarding early holistic assessment. In turn, clear advice from specialist services on the extent and scope of this assessment is needed by GPs.

Whilst transgender and gender minority patients face unacceptable years-long waits for NHS care, many seek private advice. Patients may not be well served if private assessments are less thorough; adequate communications are frequently missing and many do not have evidence that specialist follow-up is occurring. This puts transgender patients’ GPs in a difficult position of providing prescriptions and monitoring in a limbo of specialist care, and in the absence of published guidelines to fall back upon. The unsatisfactory situation presents a clear patient safety risk. The trend seen here towards more thorough monitoring at the practice with a higher prevalence of transgender patients, as well as dedicated clinics, suggests a higher risk of inconsistent care for patients at smaller practices or in areas of lower prevalence. Whilst it is incumbent on GPs to put into place systems and policies to reduce risk, it is the role of NICE and the specialist societies to develop proper evidence-based guidelines and mandated SCAs so that GPs are supported to provide safe, evidence-based care. The growing number of patients requesting gender interventions increases the urgency of embedding standardised care and recall systems.

The cumbersome way by which transgender care is managed within existing fragmented historic EHR systems in the UK is another potential contributor to poor care. Identifying a practice’s transgender patients is difficult, due to inconsistent coding, non-congruent gender markers and redacted notes. The RCGP [1] has called for improvements to EHR systems to allow for birth-registered (biological) sex as well as gender identity; Cass recommended that minors should not change NHS gender number/marker [11], not least due to the potential loss of safeguarding information. Ideally, both adult and paediatric GICs would have full seamless access to a patient’s unredacted EHR, to allow full holistic assessment and a seamless transfer and sharing of care, and to avoid the loss of important healthcare information.

The new paediatric services are expected to address wider mental needs of patients up to age 18, but it is currently unknown whether the proposed new young adult services [12] are likely to be similarly refocused, or to be modelled more closely on the more medicalised adult approach. Our data suggest that that if resources designed to support psychosocial and neurodevelopmental needs are not available as part of a holistic gender service, significant additional community resources will be needed to adequately support patients. This consideration should form part of the earliest stage of service design.

Whilst the development of guidelines, changes to the EHR and the provision of adequate support services are needed from NHS bodies, NICE and the Royal Colleges, there are specific improvements arising from this project which can be implemented at a practice level. Suggestions arising from this project include a named GP with responsibility for remaining up-to-date with changes in the field of gender medicine, an overview of processes and future re-audit; a consistent, agreed practice-wide schedule of coding for all patients presenting with gender incongruence or dysphoria; clear published local guidelines on shared care with GICs (private or NHS); the education of clinicians in the additional needs and comorbidities commonly faced by transgender patients and to be considered during assessment and review; and embedded electronic recall systems for the monitoring/health screening of shared care patients undergoing gender interventions.

### 4.6. Implications for Research

This project highlights the most important research and evidence gaps in primary care. A better long-term understanding of the risks, side-effect profiles and benefits of hormonal (and adjunctive) treatments is urgently needed to underpin standardised guidelines and monitoring, enabling GPs to be confident in the long-term care of transgender patients. An explicit analysis of the applicability of population screening programmes in transgender people is required to ensure appropriate programme coverage.

A better understanding of transgender patients, their neurodevelopmental and psychiatric presentations and long-term medical needs is required. Prospective studies will be valuable, as recommended by Cass for the paediatric population [11] as well as the outstanding data linkage study to adult GICs commissioned by her report. General practice stands in the middle of a fragmented system of gender-affirming care in the UK. Working with GICs and surgical departments, it could perform a pivotal role in research into long-term outcomes and complications. There are particular gaps in understanding the reasons for, and implications of, detransition; whilst detailed proposals for clinics are expected [12], primary care research is best placed to identify and understand the cohort, through networks such as PACT. However, whilst primary care data have been used for demographic research [22,55], concerns over the lack of clarity in data collection between sex and gender identity [61], alongside wider research failings in recording the trans status ascertainment method and the disaggregation of data by sex [57], limit and confuse study findings. The results of a UK Government-sponsored review [66] may provide much-needed guidance, with potential to influence best practice for international research in transgender healthcare. Improving current primary care EHR systems to avoid the conflation of sex and gender would be a helpful step in unlocking the potential of primary care data in transgender healthcare research.

## 5. Conclusions

This replication and extension of an original primary care QI-style audit across a new and larger general practice population in southwest England has demonstrated inadequacies in the monitoring and health screening of transgender patients. Mental health and neurodevelopmental complexity are common, adding to the challenges of supporting patients on waiting lists for NHS care or navigating the private–NHS interface. There is a demonstrable need for evidence-based primary care guidance for transgender healthcare interventions, adequately resourced and research-oriented services, improved EHR design to safely manage the complexities of a gender transition journey and embedded and consistent systems of coding and monitoring. We anticipate the results of the ongoing review of gender services may address the lack of clarity around SCAs, as clear and agreed lines of responsibility are required across all sectors to ensure transgender patients receive safe, holistic and timely care.

## Figures and Tables

**Figure 1 healthcare-13-00353-f001:**
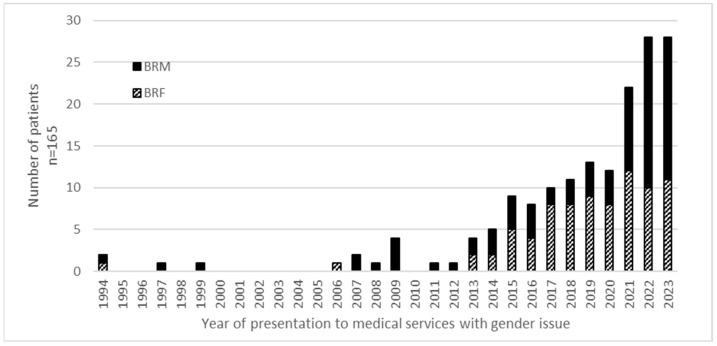
Year of first presentation to medical services with gender dysphoria/incongruence. No patients were recorded as having presented prior to 1994. All healthcare services in the UK were disrupted in 2020 due to the COVID-19 pandemic.

**Figure 2 healthcare-13-00353-f002:**
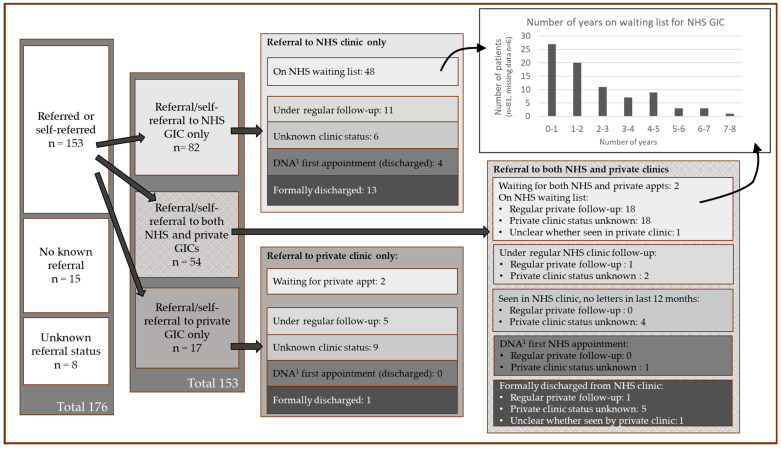
Patient care pathways and number of years patients have been waiting on a National Health Service (NHS) Gender Identity Clinic (GIC) waiting list. ‘Regular follow-up’ indicates that a letter from the clinic, dated from within the last 12 months is recorded in the patient’s notes. ‘Clinic status unknown’ indicates that a recent (<12 month) letter is not available in the notes, although the patient was known to be previously under the care of a clinic and was not formally discharged. ^1^ DNA—did not attend. NHS gender clinics generally have a policy of discharging a patient if they do not attend their first appointment.

**Figure 3 healthcare-13-00353-f003:**
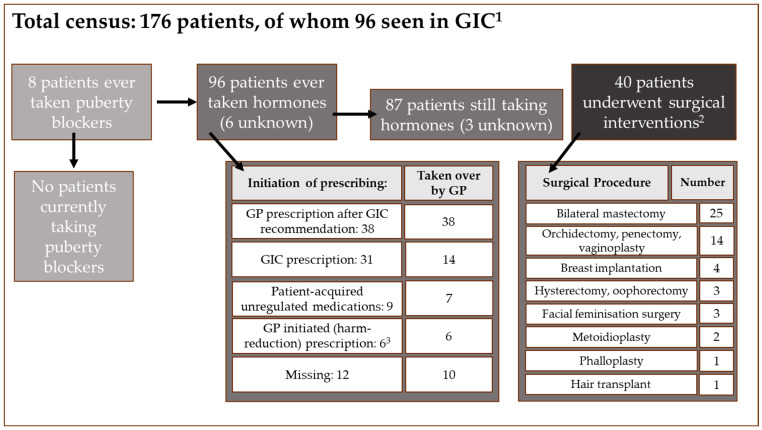
Summary of patient medical and surgical pathways, including the initiation of prescribing and surgical interventions undergone. ^1^ Two patients are recorded as having been formally discharged without having started endocrine treatments. One of these patients started hormonal treatment in an adult clinic, and the other sought surgical intervention only. ^2^ One patient was excluded as mastectomy was not for a gender-affirming indication. ^3^ An additional patient was initiated on harm-reduction hormones but stopped after GIC assessment/prescription. GIC: Gender Identity Clinic. GP: general practice.

**Table 1 healthcare-13-00353-t001:** Overview of practices involved in the project.

Practice Profile 1	Practice Profile 2
*Total population*: 42,728, serving approx. 33% of the town’s population	*Total population*: 27,614, serving approx. 78% of the town’s population
*Demographic profile* [17]: Age (years): 7% age 0–14, 44% age 15–24, 35% age 25–59 and 13% age 60+	*Demographic profile* [17]: Age (years): 14% age 0–14, 10% age 15–24, 40% age 25–59 and 36% age 60+
Deprivation index: in the 30% least deprived	Deprivation index: in the 30% least deprived
Ethnicity: 90.9% White, 2.5% Mixed, 4.3% Asian, 2.3% Non-White	Ethnicity: 97.7% White, 2.3% Non-White
*Practice policy:* 2 GPs (of 36) with special interest; 1 transgender patient clinic held/fortnight Harm-reduction hormones prescribed * where considered necessary Close links with NHS GIC and care shared with private clinics after quality assurance Patients required to continue care under GIC until formal discharge.	*Practice policy*: No specific clinician with transgender special interest No harm-reduction hormones prescribed * Care shared with NHS GIC and agreed variably with private clinics Patients required to continue care under GIC until formal discharge.

NHS: National Health Service. GIC: Gender Identity Clinic. * Harm-reduction prescribing describes the initiation of gender-affirming hormone medication, also known as ‘bridging prescriptions’, by a GP before any specialist assessment when it is considered to be in the patient’s interest. This practice is supported by the General Medical Council [8] if a doctor feels confident to do so. The Royal College of General Practitioners [1] do not consider this part of the GP’s role unless they have developed special skills.

**Table 3 healthcare-13-00353-t003:** Patient demographic data.

		Practice 1	Practice 2	Aggregated
Population	Number of TG/GM * patients	135	41	176
	Total number of patients	42,727	27,614	70,342
	Prevalence of TG/GM * patients (per 100,000 patients)	320	150	250
	Birth-registered sex female/male	51%:49% ^1^	46%:54%	50%:50% ^1^
Age at census	All: mean, median and range (in years)	26, 22, 13–70	26, 25, 14–60	26, 22, 13–70
	Birth-registered females	24, 22, 13–60 ^1^	23, 19, 16–38	23, 21, 13–60 ^1^
	Birth-registered males	29, 23, 15–70 ^1^	29, 27, 14–60	29, 24, 14–70 ^1^
Age at presentation	All: mean, median and range (in years)	21, 19, 6–62	21, 20, 12–52	21, 19, 6–62
	Birth-registered females	19, 17, 9–50 ^1^	19, 18, 12–33	19, 17, 9–50
	Birth-registered males	23, 19, 6–62 ^1^	23, 22, 12–52	23, 20, 6–62
Gender marker	Number changed % (n/N)	33% (44/134) ^2^	44% (18/41)	35% (62/176)
	Birth-registered females	31% (21/68) ^1^	47% (9/19)	34% (30/87)
	Birth-registered males	35% (23/66) ^1^	41% (9/22)	36% (32/88)

^1^ One patient of 135 excluded as patient’s birth-registered sex unknown. ^2^ Four patients had changed their gender markers to ‘Indeterminate’ (three BRF and one BRM) and two to ‘Unspecified’ (two BRF). NHS guidance recommends the use of M or F only [30]. * TG/GM: transgender and gender minority patients.

**Table 4 healthcare-13-00353-t004:** Co-morbid psychiatric/neurodevelopmental comorbidities, adverse childhood experiences and involvement by secondary mental health service amongst transgender patients.

	Number of Patients (and Missing Data) N = 176 *	% of All Patients	BRF N = 87 *	BRM N = 88 *
*Mental health diagnosis*				
Anxiety or depression	130 (14)	74%	74 (4)	56 (9)
Documented deliberate self-harm	86 (21)	49%	58 (6)	28 (14)
Eating disorder excluding ARFID ^1^	10 (29)	6%	8 (11)	2 (17)
OCD ^2^	7 (28)	4%	5 (11)	2 (16)
Personality disorder	9 (28)	5%	6 (11)	3 (16)
Psychotic or bipolar disorder	7 (32)	4%	3 (12)	4 (19)
Functional seizure disorder	8 (27)	5%	6 (10)	2 (16)
Substance misuse	47 (11)	27%	21 (4)	26 (7)
No recorded mental health diagnosis	35 (11)	20%	12 (3)	22 (8)

*Neurodevelopmental diagnosis*				
Learning difficulties	5 (8)	3%	3 (2)	2 (5)
ASD ^3^	29 (7)	16%	16 (4)	13 (2)
ADHD ^4^	22 (7)	13%	11 (4)	11 (2)
Awaiting assessment for ASD and/or ADHD	24 (14)	14%	7 (7)	17 (6)

*Other*				
Adverse childhood experiences recorded	75 (49)	43%	45 (16)	30 (32)
CAMHS ^5^ involvement	55 (52)	55%	40 (21)	15 (30)
Adult secondary care mental health team involvement ^6^	46 (24)	26%	26 (9)	20 (14)

^1^ ARFID: avoidant/restrictive food intake disorder; ^2^ OCD: obsessive-compulsive disorder; ^3^ autism spectrum disorder; ^4^ ADHD: attention deficit–hyperactivity disorder. ^5^ CAMHS: Child and Adolescent Mental Health Service. ^6^ Adult services include Community Mental Health Team (CMHT), in-patient psychiatric care, hospital liaison services and any other non-IAPT (Increasing Access to Psychological Therapies) adult secondary care mental health services. Team names vary between regions. * One patient missing from disaggregated data due to missing birth-registered sex.

**Table 5 healthcare-13-00353-t005:** Letters from NHS and private clinics.

	NHS Clinic	Private Clinic
Number of patients seen by clinic	33	52
Adequate letter in notes	25	23
Currently taking hormones	31 (0) ^1^	39 (5) ^1^

Prescribing taken over by GP	30	27
…with adequate letter	24	19
…without adequate letter	6	8

^1^ Number in brackets represents number of patients where data are missing.

**Table 6 healthcare-13-00353-t006:** Gender-affirming endocrine medications being taken.

Type of Medication	Number of Patients (Missing Data)
*GnRH* (*puberty blockers*)	
Previous	8 (15)
Current	0

*Masculinizing*	38 (3)
Topical testosterone (gel)	27
Nebido intramuscular	10
Sustanon intramuscular Medication type missing	1 0
Taking adjunct alongside testosterone ^1^	2
Previously taken, now stopped	4 (2)


*Feminizing*	49 (2)
Topical oestrogen only (gel/patch/spray)	25
Oral oestrogen only	20
Medication type missing Oestrogen of any form and adjunct ^1,2^ Previously taken, now stopped	4 23 (3) 2 (1)

^1^ GP prescriptions for adjuncts were the GnRHs triptorelin and leuprorelin; private clinics also included spironolactone and finasteride. ^2^ Two patients known to be taking self-acquired cyproterone acetate.

**Table 7 healthcare-13-00353-t007:** Monitoring of patients using hormonal medication against audit standard.

Within the Last 12 Months	Masculinising Hormones N = 38, n (%)	Feminising Hormones N = 49, n (%)
Oestradiol	*n/a*	36 (73)
Testosterone	34 (89)	34 (69)
FBC/HCT	34 (89)	*n/a*
LFTs	32 (84)	31 (63)
Lipids	31 (82)	36 (73)
HbA1C	18 (47)	*n/a*
Prolactin	*n/a*	29 (59)
**Subtotal blood monitoring**	**18 (47)**	**21 (43)**
Blood pressure	21 (55)	24 (49)
BMI or weight	19 (50)	19 (39)
**All annual monitoring**	**7 (18)**	**7 (14)**

**Table 8 healthcare-13-00353-t008:** Audit of population screening reminders.

Anatomy-Appropriate Reminder in Notes Where Gender Marker Had Been Changed	BRF	BRM
Cervical screening	6/28 (21%)	
Breast screening	1/14 (7%)	
Presence of prostate noted		4/32 (13%)
AAA screening		1/32 (3%)

## Data Availability

Data available on request due to privacy restrictions. The data presented in this study are available on request from the corresponding author. The data are not publicly available due to patient confidentiality.

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
