# Peer review of "Towards Best-Practice Healthcare for Transgender Patients: Quality Improvement in United Kingdom General Practice"

_healthcare, 2025, doi:10.3390/healthcare13040353_

Round 1

Reviewer 1 Report

Comments and Suggestions for Authors

This paper is of clinical significance in the UK and globally.  However, due to the current position in the UK, its relevance is in the UK only. 

I have uncertainty regarding the comparison audit between the practices due to the differences in demographics.  I do wonder if an audit of one practice would have been better placed and then conclusions drawn from that.  Then further comparison audits undertaken.

I acknowledge the footnote regarding language and whilst gender dysphoria is still used (and is the WHO term), gender incongruence is the term preferred by the trans community and is taught as the correct term in healthcare education in the UK - please could you consider this inclusion.

Author Response

Dear Reviewer 1 - many thanks for your constructive feedback which we have addressed as follows.

Comment 1: This paper is of clinical significance in the UK and globally.  However, due to the current position in the UK, its relevance is in the UK only. 

Response 1: Thankyou for your comment, with which we agree. I have substantially added to Section 4.1  regarding limitations re relevance of audit standard and clinical results to UK only (Lines 733-740, p20) However, we maintain that the key findings should be transferable to other global health care systems, and that we hope that the approach here might therefore provide a template for necessary future research in other health care settings.

Comment 2: I have uncertainty regarding the comparison audit between the practices due to the differences in demographics.  I do wonder if an audit of one practice would have been better placed and then conclusions drawn from that.  Then further comparison audits undertaken.

Response 2: Thankyou for this feedback. The study grew partly due to staff synergies between practices, and it was also felt that the differences in demographics and approach to transgender care may help strengthen the audit results by presenting useful differences in demographic characteristics, pathways and audit results which could be used to share lessons and improve local services. We have added a sentence (lines 132-134, p3) to clarify this. 

Comment 3: I acknowledge the footnote regarding language and whilst gender dysphoria is still used (and is the WHO term), gender incongruence is the term preferred by the trans community and is taught as the correct term in healthcare education in the UK - please could you consider this inclusion.

Response 3: Many thanks for your comment. We have added a Box (Box 1, Line 95-96, p3) explaining why the term gender dysphoria was used. 

Many thanks - Carine, on behalf of the authors

Reviewer 2 Report

Comments and Suggestions for Authors

Dear authors, the presented study addresses a very important question about improving care for transgender people in the UK health system.
However, in general, I feel that the development of the content of the manuscript needs to be substantially improved in order to more clearly identify the methodology employed. As I understand it, this is an observational study based on medical records. For observational studies, the STROBE guideline, which includes the items to be included in an observational study report, should be followed.

The title should identify the study design as observational.

The abstract should be more informative with regard to the methodology followed (design, population and setting, inclusion and exclusion criteria, variables, analysis process, ethical issues.

Introduction: is adequate in content and length. The acronym NHSE on line 58 (page 2) should be revised and corrected. End the introduction with the objective of the study (do not add the ‘note on language’ at the end of the introduction; this information should be addressed earlier in the introduction).

Materials and methods: Start by describing the research design. This is a multicentre study, but this issue needs to be better clarified before describing the characteristics of the individual centres in table 1.
This information should be reported in the participants section. For each of the centres, the estimated sample size calculation for the study should be estimated so that there is statistical significance.

The first paragraph of subsection 2.2 contains information that corresponds to the introduction. this information should be addressed as background information of interest to contextualise and justify the study.

Inclusion and exclusion criteria for the study (dates of records, age of patients...) are not explained clearly enough.

The statistical analysis section should be explained more clearly (do not start with the missing data, but how the data in the records will be analysed).

Results: The results should follow a logical order in which first the socio-demographic results are described, then the descriptive results of the sample and finally the inferential results. It is not correct to state in the first sentence of the results that ‘this study found significant variability’ when previously, in the methodology, the authors indicated: ‘No tests for statistical significance were used’.

The results should not contain citations to other studies, which should be addressed later in the discussion.

Statistical data should be better organised and highlighted in table 3 to avoid confusing the reader in their interpretation.

In my opinion, Figure 1, 2 and 3 are irrelevant (this information can be described in the text of the manuscript).

References: the style should be checked and a link to the primary source or DOI should be added where appropriate.

Author Response

Dear Reviewer 2

Many thanks indeed for your helpful comments. We have addressed these as set out below.

Comment 1: As I understand it, this is an observational study based on medical records. For observational studies, the STROBE guideline, which includes the items to be included in an observational study report, should be followed. The title should identify the study design as observational.

Response 1: Thankyou for giving us the opportunity to clarify. This is quality improvement/audit work, not an observational study.   Therefore it was not appropriate to follow the STROBE guidance and we used the words ‘quality improvement’ in the title. Please note that we have reorganised the Methods (p3-7), as per your suggestions, to improve clarity around the study design.

Comment 2: The abstract should be more informative with regard to the methodology followed (design, population and setting, inclusion and exclusion criteria, variables, analysis process, ethical issues.

Response 2: Many thanks. Additions have been made to the abstract regarding setting, study design, ethics (see lines 11-13,17,18; p1).

Comment 3: Introduction: is adequate in content and length. 

Response 3: Many thanks. 

Comment 4: The acronym NHSE on line 58 (page 2) should be revised and corrected. 

Response 4: Thankyou, this has been corrected (line 65, p2).

Comment 5: End the introduction with the objective of the study (do not add the ‘note on language’ at the end of the introduction; this information should be addressed earlier in the introduction).

Response 5: Thank you for this advice, we have altered the end of the introduction (lines 106-113, p3) to better clarify the objectives, and we have moved this information to regarding use of SAGER guidelines to Section 2.6 (lines 273-278, p7). Clarification regarding use of language has been added to the new Box 1 (line 95-96, p2).

Comment 6: Materials and methods: Start by describing the research design. This is a multicentre study, but this issue needs to be better clarified before describing the characteristics of the individual centres in table 1.

Response 6: Thank you, we have made the suggested changes (line 125, p3).

Comment 7: This information should be reported in the participants section. For each of the centres, the estimated sample size calculation for the study should be estimated so that there is statistical significance.

Response 7: The study was not designed to be powered to report on statistical significance. We report descriptive data, and tentatively describe trends in that data (see line 225-226 (p6), 271-273(p7))

Comment 8: The first paragraph of subsection 2.2 contains information that corresponds to the introduction. this information should be addressed as background information of interest to contextualise and justify the study.

Response 8: Thankyou for this comment. We have reorganised this section into its appropriate parts - the first sentence has been moved to the Introduction (lines 75-78, p2), and the rest (as it discussion of the results of the work) in the discussion section (4.1, lines 702-744, p19-20).  

Comment 9: Inclusion and exclusion criteria for the study (dates of records, age of patients...) are not explained clearly enough.

Response 9: Many thanks. We have reworded and added to this paragraph (line 228-243, p6) and also added limitations to the search strategy in the limitations section (lines 967-981, p24).

Comment 10: The statistical analysis section should be explained more clearly (do not start with the missing data, but how the data in the records will be analysed).

Response 10: Many thanks. We have reordered this section (lines 271-289, p7) and moved the limitations regarding missing data to the limitations section (section 4.3, lines 967-981, p24). We’ve added clarification that we were not powered for statistical analyses (lines 225-226 (p6), 271-273(p7))

Comment 11: Results: The results should follow a logical order in which first the socio-demographic results are described, then the descriptive results of the sample and finally the inferential results. 

Response 11: We have reorganised the results  and discussion (e.g. much of Section 3.1 p7-10 has been moved to Section 4.1 (p19-21); Sections 4.2 (p21-23) and 4.3 (p24-25) have been reordered to reflect presentation of the results); thank you for the constructive feedback which has helped us improved the clarity of the writing. 

Comment 12: It is not correct to state in the first sentence of the results that ‘this study found significant variability’ when previously, in the methodology, the authors indicated: ‘No tests for statistical significance were used’.

Response 12: Thankyou for pointing this out. 'Significant' was (incorrectly) used as a synonym for substantial, which has now replaced it in the text (line 329, p8).

Comment 13: The results should not contain citations to other studies, which should be addressed later in the discussion.

Response 13: Again, thankyou for this helpful suggestion. We have moved the reference-heavy section 3.1 (lines 298-325, p7-8 and lines 376-428, p9) on guideline comparison to the Discussion (section 4.1, lines 702-806, p19-21), and summarised how the audit standard was created in the Results section (lines 327-372, p8) and so that it does not require citations.

Comment 14: In my opinion, Figure 1, 2 and 3 are irrelevant (this information can be described in the text of the manuscript).

Response 14: Thankyou for the feedback. We have removed Figure 1 and added details into the text (lines 458-461, p11). Previous Figure 2 (now Figure 1, line 476, p12) has been retained as the increase in referral rate is illustrated better with a graph than numerically described, and it mirrors similar presentation of data in other literatures (e.g. the Cass Report). Figure 3 has been incorporated into the new Figure 2 (line 511, p13). 

Comment 15: References: the style should be checked and a link to the primary source or DOI should be added where appropriate.

Response 15: Many thanks - we have checked through the references and added the missing DOIs to where available, a PMID  where no DOI available [4], an online link [33], and made the DOI citation style more consistent for the rest of the references .

Many thanks, Carine, on behalf of the authors

Reviewer 3 Report

Comments and Suggestions for Authors

Introduction and Objectives

The introduction provides a solid context regarding the challenges of transgender healthcare. However, the objectives could be more explicitly stated. A clear articulation of the study's purpose and its expected contribution to current practices would improve focus.

Structure and Cohesion

While the manuscript is generally well-organized, some sections, particularly the transition between results and discussions, lack fluidity. Reorganizing these parts to better align findings with their implications could improve readability.

Methodological Rigor

The methodology is robust but underexplained in parts. The rationale for selecting audit standards should be detailed, particularly how these align with local and international guidelines. Additionally, the discussion of potential biases or limitations in data collection and interpretation is underdeveloped.

Data Presentation

While the data is comprehensive, its presentation could benefit from additional visual aids. Graphs or flowcharts summarizing the patient care pathways and key findings would enhance accessibility and engagement. Highlighting significant trends or anomalies would further aid comprehension.

Engagement with Literature

The manuscript engages adequately with existing literature but misses critical opportunities for deeper comparisons with international standards. Incorporating recent studies and exploring their alignment or divergence with the findings would strengthen the manuscript’s scholarly impact.

Recommendations

The recommendations are broad and would benefit from more specificity. Practical, actionable strategies for policymakers and healthcare providers, as well as suggestions for integrating improved EHR systems, should be emphasized. The ethical implications of shared care agreements also warrant further discussion.

Language and Sensitivity

The manuscript demonstrates a professional tone but occasionally uses complex sentence structures that reduce clarity. Simplifying language without sacrificing academic rigor is advised. Consistent terminology, particularly when referencing gender and healthcare practices, would reduce ambiguity.

Author Response

Dear Reviewer 3

Many thanks for your helpful and constructive comments which we have addressed as set out below.

Comment 1: Introduction and Objectives: The introduction provides a solid context regarding the challenges of transgender healthcare. However, the objectives could be more explicitly stated. A clear articulation of the study's purpose and its expected contribution to current practices would improve focus.

Response 1: Thankyou for this feedback. The aims of the study have been reworded in the last sentence of the introduction and are hopefully now clearer (lines 106-113, p3).

Comment 2: Structure and Cohesion: While the manuscript is generally well-organized, some sections, particularly the transition between results and discussions, lack fluidity. Reorganizing these parts to better align findings with their implications could improve readability.

Response 2: Many thanks. We have reorganised the results  and discussion (e.g. much of Section 3.1 (p7-10) has been moved to Section 4.1 (p19-21); Sections 4.2 (p21-23) and 4.3 (p24-25) have been reordered to reflect presentation of the results); We have reorganised the results and discussion, such that the discussion underpinning the creation of the audit standard has been moved to a deeper discussion of guidelines (in new section 4.1) and the order of the Discussion mirrors the order of the Results. We hope this creates better flow, and links the limitations in guidelines with the audit results.

Comment 3: Methodological Rigor: The methodology is robust but underexplained in parts. The rationale for selecting audit standards should be detailed, particularly how these align with local and international guidelines. 

Response 3: We have expanded on how guidelines were selected in Section 4.1 (686-744, p19-21) including the limitations of our approach.

Comment 4: Additionally, the discussion of potential biases or limitations in data collection and interpretation is underdeveloped.

Response 4: Thankyou. We have expanded on the limitations of the populations pool (lines 982-988, p24-25) and the possible biases in interpretation (1000-1004, p25).

Comment 5: Data Presentation: While the data is comprehensive, its presentation could benefit from additional visual aids. Graphs or flowcharts summarizing the patient care pathways and key findings would enhance accessibility and engagement. Highlighting significant trends or anomalies would further aid comprehension.

Response 5: Thankyou for your suggestion. We have added two figures which I hope are helpful for the visualisation of patient pathways, and  edited the text in order to prevent too much repetition (Lines 511, p13 and line 556, p15). 

Comment 6: Engagement with Literature: The manuscript engages adequately with existing literature but misses critical opportunities for deeper comparisons with international standards. Incorporating recent studies and exploring their alignment or divergence with the findings would strengthen the manuscript’s scholarly impact.

Response 6: Many thanks. We have expanded on the comparison and limitations of international guidance to the UK situation in section 4.1 (lines 686-744, p19-20), including two further examples of international guidance [30,45]. Comparison with relevant results of a very recent study, published during this revision process [54] have been included (lines 812-814, p21) as well as conclusions from an recent international systmatic review [64] (lines 852-5 (p22) and 1088-9 (p26)). 

Comment 7: Recommendations: The recommendations are broad and would benefit from more specificity. Practical, actionable strategies for policymakers and healthcare providers, as well as suggestions for integrating improved EHR systems, should be emphasized. The ethical implications of shared care agreements also warrant further discussion.

Response 7: Many thanks. We have clarified suggested improvement in this section, both at GP practice (lines 1057-64, p26) and policy levels (lines 1013-4,  1028-1033, p25) and we have emphasised the need for shared EHR systems (lines 1040-1043, p25-6). We have discussed the difficulties with shared care in both the introduction and in the implications section; we look forward to the results of the ongoing Adult Gender Services review and anticipate that this may further influence implications in this area, and have included a sentence to this effect in the Conclusion (lines 1104-1106, 27). 

Comment 8: Language and Sensitivity: The manuscript demonstrates a professional tone but occasionally uses complex sentence structures that reduce clarity. Simplifying language without sacrificing academic rigor is advised. Consistent terminology, particularly when referencing gender and healthcare practices, would reduce ambiguity.

Response 8: Thank you, we have revised the complex sentences to improve clarity e.g. lines 1048-1051 (p26), 1024-1028 (p25), 1104-1106 (p27).

Many thanks, Carine, on behalf of the authors

Round 2

Reviewer 2 Report

Comments and Suggestions for Authors

Dear authors, thank you very much for the clarifications and justified responses to the previous round of revisions. The manuscript has been substantially improved in the new version.

Just a few minor issues:

Although the acronym UK is internationally known to refer to United Kingdom, I recommend not to use the acronym in the title and to explain it in its first use in the abstract (in fact, the authors explain the acronym in the first sentence of the introduction).

Introduction: the information in box 1 is more correct in the text of the introduction without being in the box.

The results and discussion section have been substantially improved and bring more coherence and clarity to the manuscript. Although the results are not statistically significant, they are of interest due to the relevance of the object of study.

Author Response

Dear Reviewer 2

Thankyou again for the further comments, which we addressed as follows: 

Comment 1: Thank you very much for the clarifications and justified responses to the previous round of revisions. The manuscript has been substantially improved in the new version.

Response 1: Many thanks, and also for your previous comments which were very helpful in improving the manuscript. 

Comment 2: Although the acronym UK is internationally known to refer to United Kingdom, I recommend not to use the acronym in the title and to explain it in its first use in the abstract (in fact, the authors explain the acronym in the first sentence of the introduction).

Response 2: Many thanks for this feedback. We have changed the acronym to the full term United Kingdom in the title (line 3, p1), and explained the acronym in the first line of the abstract (line 9, p1).

Comment 3: Introduction: the information in box 1 is more correct in the text of the introduction without being in the box.

Response 3: Thankyou, we have moved the information in the box into lines 104-111, p3 of the introduction. 

Comment 4: The results and discussion section have been substantially improved and bring more coherence and clarity to the manuscript. Although the results are not statistically significant, they are of interest due to the relevance of the object of study.

Response 4: Many thanks, we agree that the results may be useful to other clinicians in this evolving area of healthcare. 

With thanks and kind regards, Carine (on behalf of the authors)